# The *Privacy Onion Effect*: Memorization is Relative

**Nicholas Carlini**          **Matthew Jagielski**          **Nicolas Papernot**
**Andreas Terzis**            **Florian Tramer**             **Chiyuan Zhang**
*Google Research*

## Abstract

Machine learning models trained on private datasets have been shown to leak their private data. While recent work has found that the *average* data point is rarely leaked, the *outlier* samples are frequently subject to memorization and, consequently, privacy leakage. We demonstrate and analyse an *Onion Effect* of memorization: removing the "layer" of outlier points that are most vulnerable to a privacy attack exposes a new layer of previously-safe points to the same attack. We perform several experiments to study this effect, and understand why it occurs. The existence of this effect has various consequences. For example, it suggests that proposals to defend against memorization without training with rigorous privacy guarantees are unlikely to be effective. Further, it suggests that privacy-enhancing technologies such as machine unlearning could actually harm the privacy of other users.

## 1 Introduction

Deep learning models have been shown to memorize their training data [7, 8, 11, 29, 35], and this property is often a privacy violation when the training data is sensitive [8]. This risk incentivizes the removal of private data from a model's training set in a number of scenarios. For example, a user might withdraw consent to use their data for learning, if they deem the privacy risk to be too high. Such removal interventions are increasingly formalized in legislation (e.g., the GDPR or CCPA). Alternatively, the party that trains a model (e.g., a company) may wish to proactively assess the privacy risk [16, 23] of different points in the training set, in order to adaptively remove the data points whose privacy is most at risk. Indeed, prior work has shown that the privacy of training data is highly non-uniform: while data points tend to be well protected *on average*, the empirical risk of privacy leakage (resulting from attacks) is concentrated on a small fraction of *data outliers* [2, 6, 12, 21, 33].

We show that removing easy-to-attack training examples—i.e., those most at risk from privacy attacks—has counter-intuitive consequences: by removing the most vulnerable data under a specific privacy attack and retraining a model on only the previously safe data, a new set of examples in turn becomes vulnerable to the same privacy attack. We call this phenomenon the *Privacy Onion Effect*, which we define as follows:

> *Removing the "layer" of outlier points that are most vulnerable to a privacy attack exposes a new layer of previously-safe points to the same attack.*

In this paper, we consider a canonical family of privacy attacks called *membership inference attacks* [14], which predict whether or not a given example is contained in the model's training set [29]. Membership inference attacks are the most commonly used privacy metric due to their simplicity [6, 34], generality across domains [21, 22, 27, 37], and utility as the basis for more sophisticated attacks [8]. We empirically corroborate this Privacy Onion Effect on standard neural network models trained on the CIFAR-10 and CIFAR-100 image classification datasets [19]. For example, we find that if we remove the 5,000 training samples that are most at risk from membership inference, in the

36th Conference on Neural Information Processing Systems (NeurIPS 2022).

absence of any other effects we should mathematically expect this removal to improve the overall privacy by a factor of $15\times$, but in reality it only improves privacy by a factor of $2\times$. That is, the Privacy Onion Effect has caused this removal to be over $6\times$ less effective than expected.

We perform several experiments that refute various potential explanations for the presence of this Onion Effect—for example, we find that the effect is not explained by statistical noise, by the reduction in the training set's size, by the presence of duplicate training examples, or by the model's limited capacity. Our experiments however suggest that the Onion Effect may be explained by inliers that become outliers when more extreme outliers are removed.

The Privacy Onion Effect has significant consequences for commonly-used empirical approaches to data privacy and custody:

- **Current privacy auditing is unstable**: It is an increasingly common practice to empirically measure—or *audit*—the privacy risk of users, by instantiating concrete attacks such as membership inference [16, 23]. Our work shows that such privacy auditing lacks "stability", as a user's empirical privacy risk can vary significantly due to the removal of a small fraction of training data (a similar effect has been shown under adversarial *additions* to the data [33]).

  In the future, privacy audits should be dynamically updated as the underlying training data changes. Yet even then, an audit that reveals a low risk to attacks might provide users with a *false sense of security*, as the risk could drastically increase after the removal of a few other users' data.

- **Machine unlearning can degrade others' privacy**: Users whose privacy appears most at risk (e.g., as revealed by a privacy audit) may be the most likely to proactively request for their data to be removed from a model's training set. As we show in Section 5.2, by honoring such *machine unlearning* requests [5], a model provider might inadvertently degrade the privacy of other users.

## 2   Related Work

**Memorization and differential privacy.**   While some forms of memorization may be desirable for learning to succeed, we focus on *unintended* memorization. In this context, Feldman introduces a simple definition of memorization [11]: informally, a model memorizes a training example's *label* if removing this example from the training set significantly changes the model's probability of outputting this label. Such label memorization is provably prevented by training models with differential privacy (DP) [1, 10], as DP guarantees that the model's outputs are not highly affected by the addition or removal of any training example.

Training with strong DP guarantees typically comes at a cost in accuracy [32]. Prior work offers an explanation for this tension between privacy and accuracy [4, 11], by showing that memorization is necessary for high-accuracy learning on certain data distributions. Examples of such distributions are those with long tails [11], and have been explored in several empirical studies [2, 12, 30].

These studies point to the worst-case nature of the differential privacy guarantee: DP provides a uniform guarantee that applies to *any* dataset. Instead, our work is motivated by the question of whether memorization can be prevented using privacy mechanisms that are tailored to the data points that are at risk (as such, our work relates to the broad literature on *instance-specific* differential privacy [25]). Specifically, we investigate if removing points that are easily memorized precludes the need to uniformly bound the privacy leakage of the training algorithm for all possible datasets.

**Membership inference.**   Perhaps the most studied attack against privacy, membership inference [14], considers an adversary that aims to answer the following question: was a given example part of a training dataset? In the machine learning setting [29], the membership inference adversary is typically given access to a model's predictions with varying granularity [24, 28, 34], ranging from the full confidence vector to the label of the class with the largest confidence score [9]. In this work, we use a specific membership inference attack, the Likelihood Ratio Attack (LiRA) [6] as described in Section 3.1, because it achieves state-of-the-art attack performance across all metrics.

# 3 The Privacy Onion Effect

We now provide experimental evidence for the Privacy Onion Effect described in the introduction: by removing the first "layer" of easy-to-attack training examples and retraining the model, we expose a new "layer" of training examples to the same attack.

## 3.1 Notation and Problem Setup

**Notation.** Let $X = \{(x_i, y_i)\}_{i=1}^{N}$ denote a training dataset; in this paper we focus on the CIFAR-10 and CIFAR-100 datasets [19], each with 50,000 labeled training examples. We use the notation $\mathcal{T}(X)$ to denote the distribution of models we would obtain by training a neural network on the dataset $X$. For any set $s \in 2^{[N]}$, we use the notation $X_s \equiv \{(x_i, y_i) : i \in s\}$ to denote a subset $X_s \subset X$. Let $\mathcal{A}(x, f)$ denote the result of running a membership inference attack on the model $f$ and example $x$. For example, for a model $f \leftarrow \mathcal{T}(X_s)$ trained on $X_s$, a good attack would predict $\mathcal{A}(x, f) = 1$ if and only if $x \in X_s$, and would return 0 otherwise.

**The Likelihood Ratio Attack (LiRA).** Given a machine learning model $f^* \leftarrow \mathcal{T}(X_{s^*})$ trained on a dataset $X_{s^*} \subset X$, LiRA first trains multiple "shadow models" $f_s \leftarrow \mathcal{T}(X_s)$ on random subsets of $X_s \subset X$. For a target example $x \in X$, LiRA then computes the logit-gap (the difference between highest and second-highest logit) $\mathcal{L}(x, f_s^{\text{in}})$ for shadow models $f_s^{\text{in}}$ that were trained on $x$, and the logit-gap $\mathcal{L}(x, f_s^{\text{out}})$ for shadow models $f_s^{\text{out}}$ that were not trained on $x$. Both distributions of logit-gaps are modeled as univariate Gaussians. Finally, to predict whether the example $x$ is contained in the training set $X_{s^*}$ of the model $f^*$, LiRA computes the logit-gap $\mathcal{L}(x, f^*)$ and compares the likelihood of this observed value under the two Gaussian distributions above. Whichever is more likely determines if $x$ was a member or not.

**Computing privacy scores.** Given a training dataset $X$ we can compute a *privacy score* for each example $x \in X$ by measuring the average *Attack Success Rate* (ASR) of our membership inference attack. We first fix a distribution $\mathbb{D}_X$ over subsets of $X$ (we consider the distribution obtained by picking each example $x \in X$ independently with probability 50%). We then randomly sample a subset $X_s \leftarrow \mathbb{D}_X$, train a model $f_s \leftarrow \mathcal{T}(X_s)$ on this subset, and compute the average attack success rate of an example $x \in X$ as the probability (over the random choice of subset, and the training randomness) that the attack correctly predicts whether or not $x \in X_s$. Formally, we denote this by:

$$\text{ASR}(x, X) \coloneqq \Pr_{f_s \leftarrow \mathcal{T}(X_s), X_s \leftarrow \mathbb{D}_X} \big[ \mathcal{A}(x, f_s) = \mathbb{1}[x \in X_s] \big].$$

We will sometimes also refer to the *advantage* of the attack (over random guessing), which is defined as $2 \cdot \text{ASR} - 1$ (i.e., an attack with a success rate of 50% has an advantage of 0).

**Plotting membership inference attack success rates.** To visualize the performance of a membership inference attack, we plot a Receiver Operating Characteristic (ROC) curve which compares the attack's true-positive rate (TPR) and false-positive rate (FPR). A trivial random guessing will achieve a TPR equal to the FPR. Stronger attacks appear further up and to the left having higher TPRs for lower FPRs. ROC evaluations are consistent with established best practices [6].

## 3.2 Our Main Experiment

Our main experimental methodology follows a simple three step process.

**Compute a privacy score for each training example.** Using the LiRA membership inference attack (described above) we begin by measuring the average attack success rate (ASR) for each example in the training dataset $x \in X$. To ensure statistical validity of our results, we compute this average across 200,000 models. We use an efficient open-source training pipeline [20] to train each model in just 16 GPU-seconds (we train on 16 A100 GPUs for a total of 1000 GPU-hours). We then run LiRA on each of these models to obtain the attack success rate of every example in the original dataset.

**Remove the least private examples.** These privacy scores computed above allow us to sort all examples based on how easy (or hard) they are to attack. We then remove the 5,000 most-vulnerable examples from the dataset (10% of the dataset overall for both CIFAR-10 and CIFAR-100). Consistent with previous observations, the most vulnerable examples are generally outliers, such as atypical or

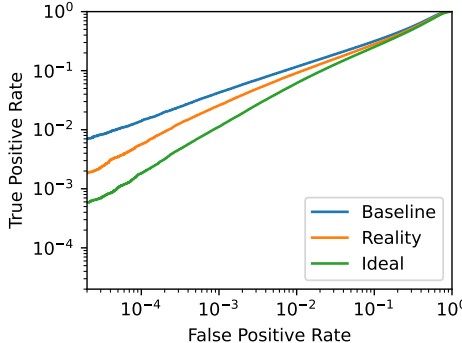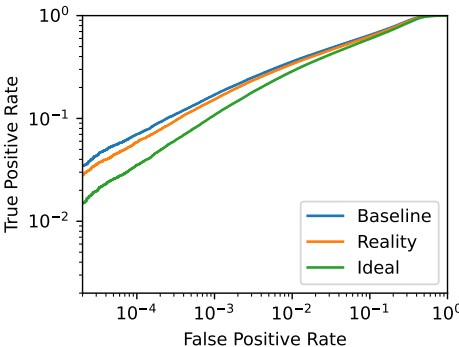

Figure 1: ROC curve when performing a baseline membership inference attack (blue) on CIFAR-10 (left) or CIFAR-100 (right), by comparing the attack's true-positive rate to its false-positive rate. Artificially preventing the attack from attacking the $5{,}000$ least private examples gives a new (green) curve; by doing this the attack now succeeds $15\times$ and $5\times$ less often for CIFAR-10 and CIFAR-100, respectively. However if we actually remove the $5{,}000$ least private examples and re-run the experiment (orange curve), we find this does not significantly improve privacy: it is $6\times$ less effective than we would have expected. This suggests that simply removing the current at-risk outliers will cause new examples to become outliers, and not solve the problem completely.

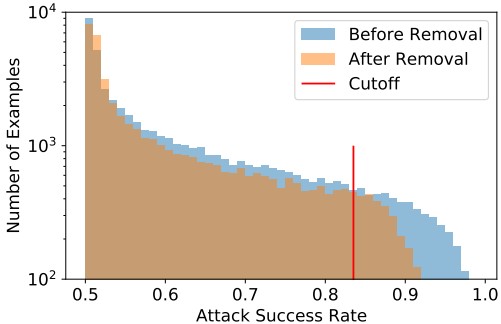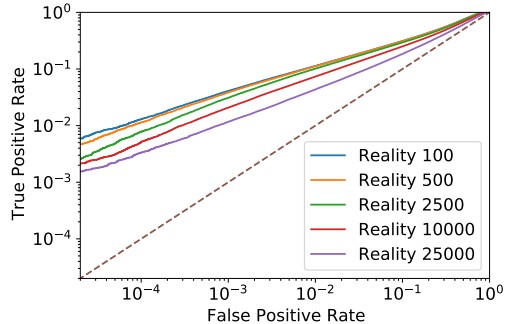

Figure 2: The Onion Effect shown as a histogram of per-example success rates. We remove all examples with attack success rates larger than the red line. In the Onion Effect, examples which remain after removal have their attack success rates increase past the red line.

Figure 3: While we primarily investigate the Onion Effect with 5000 removed examples, removing even up to the 25000 most vulnerable examples does not significantly reduce vulnerability.

sometimes even mislabeled examples. Figure 10 in the Appendix visualizes the removed samples that are easiest to attack—as we can see, these examples match the outlier examples identified by [12]. This gives a new dataset of $45{,}000$ "hard-to-attack" examples.

**Re-compute the privacy scores of each example.** Finally, we repeat the first step and train a new set of models on this modified, smaller dataset of "hard-to-attack" examples. The models trained on the reduced-size dataset have at most two percentage points lower accuracy than the baseline models trained on the full dataset. Using these models, we then re-run LiRA and measure the average attack success rate on the retained $45{,}000$ points. In an ideal environment with no other confounding effects, we would predict these points to be no-easier to attack than they were in our baseline experiment, where we used the entire training set.

### 3.3 Results

Does removing the easy-to-attack examples significantly improve privacy? Surprisingly, we find the answer is no!

Figure 1 presents the main results of our analysis with three ROC curves when evaluated on CIFAR-10 (left) and CIFAR-100 (right). The uppermost curve (in blue) plots the baseline ROC for LiRA when attacking the full 50,000 example dataset. (This curve is consistent with prior work [6] when attacking CIFAR-10 and CIFAR-100.) Then, the lowermost ROC curve (in green) shows the **ideal setting** where the attacker is limited to inferring membership for just the 45,000 least-vulnerable examples. That is, the adversary's guesses on the 5,000 easiest-to-attack examples are ignored when computing the attack success rate. This curve represents the attack performance (conversely, the privacy) we would *expect* to obtain in an idealized environment after removing the 5,000 most vulnerable outlier points—in the absence of any other effects.

However, **as the main result of our paper**, we find that this idealized setting is wrong: when we actually remove the 5,000 outliers, retrain a model, and re-run the membership inference attack, it is much easier to attack the remaining 45,000 examples than we predicted in the idealized setting. This is shown by the middle ROC curve (in orange) in Figure 1. Note that the y-axis is on a log-scale, and thus the difference between the idealized result and the obtained result is approximately half an order of magnitude. Concretely, at a fixed false-positive rate of 0.01%, the baseline true-positive rate is 1.5%. By restricting the attack to targeting the 45,000 most private samples, the idealized true-positive rate would drop to 0.1%. (To compute this value, we perform a membership inference attack on every example in the dataset and select a threshold that yields a false-positive rate of 0.01%.) Then, we compute the true-positive rate of the attack averaged across every example in the dataset *except for the 5,000 most vulnerable examples we will remove*. We find that the attack has a TPR of 0.1% on the remaining 45,000 examples.

However, when we actually run this experiment (i.e., we drop the 5,000 most vulnerable examples from the training set and retrain the model), the attack's TPR on the remaining 45,000 examples only drops by roughly half, to 0.6%. Removing outliers is thus over 6× less effective at mitigating membership inference than we would have expected. We also present this effect as a histogram of per-example attack success rates in Figure 2.

As a brief aside, a natural generalisation of the above removal procedure is to iteratively and adaptively remove smaller numbers of examples, rather than 5,000 all at once. In the Appendix we show this procedure does not significantly impact our findings, and so for simplicity consider one-shot removal for the rest of this paper.

**The remainder of this paper** investigates the question: *why does removing the least private examples not result in a privacy-preserving model?* We provide evidence of an *Onion Effect* that explains this surprising phenomenon: the least private examples are "outliers" which, when removed, expose a new set of examples to become outliers—these new outliers are then (nearly) as vulnerable as the previous outliers. In consequence, removing examples that are at risk of privacy attacks does little to solve the privacy problem—we just shift the issue onto a new set of outliers.

**An illustrative example in SVMs.**   To provide intuition for this Onion Effect, it may be illustrative to consider the special case of support vector machines (SVMs), where this effect appears naturally and explicitly. SVMs are defined by a set of *support vectors*—training examples that lie close to the model's decision boundary. When trained without explicit privacy guarantees, an SVM's decision boundary thus leaks information about these specific training examples. Yet, if we removed these support vectors from the training set (the first "onion layer") and retrained the SVM, a new set of examples would now lie closest to the model's decision boundary, and these examples will thus be selected as support vectors and be at risk (the second "onion layer"). While deep neural networks are very different than SVMs, this paper shows neural networks exhibit a similar empirical phenomenon.

## 4   Potential (Incorrect) Explanations for the Onion Effect

In this section we consider various hypotheses that might explain the Onion Effect, but upon further investigation are not correct. We focus exclusively on the CIFAR-10 dataset due to the computational cost associated with the experiments we perform (this cost is primarily due to the large number of models that need to be trained).

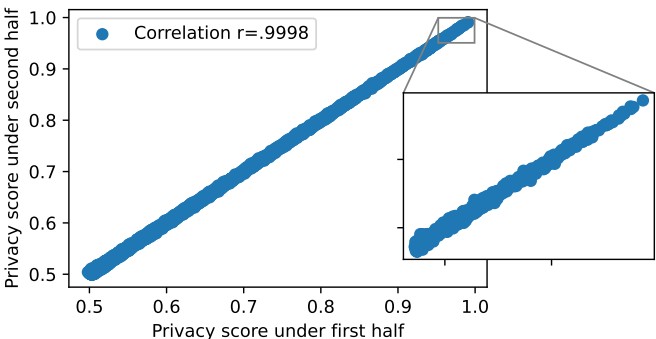

Figure 4: The Onion Effect is not due to statistical noise. Our privacy metric is stable across two independent runs and has an almost perfect correlation (.9998).

## 4.1 The Onion Effect Is Not Explained by Statistical Noise in the Privacy Metric

It is possible that the Onion Effect could be due to statistical noise in the measurement of privacy scores. That is, after removing the (what we believe to be) easiest-to-attack examples, a new set of examples would "become" the new easy-to-attack simply due to high variance in the privacy scores. We find this is not the case.

Suppose that the privacy measurement of each example consisted of two components

$$\mathrm{priv}_{\mathrm{observed}}(x) \coloneqq \mathrm{priv}_{\mathrm{true}}(x) + \mathrm{noise}(x)$$

where the first factor is the "true" privacy score for this example, and the second factor is some random experimental noise. Even if the total magnitude of this noise is small, if many values have very similar (true) privacy scores, then the $5,000$ least private examples that we remove might significantly change from one experimental run to the next. Therefore, when we remove these examples, we might not actually be removing the truly least private examples, but a (somewhat) random subset of the least private examples. This could explain our observation: when we remove these examples, and run our experiment again, some of the remaining examples will have higher privacy scores by chance alone. To refute this hypothesis, we show that while there does exist *some* experimental noise, the total magnitude of this noise is negligible and cannot by itself explain the Onion Effect.

In Figure 4, we scatter plot each example's average attack success rate when computed on a set of $100,000$ models, against that example's average attack success rate when computed over another, independent set of $100,000$ models. The correlation in the scores is near-perfect ($r = 0.9998$), indicating that noise is not the primary cause for the Onion Effect. We also include a pane to zoom in on the right hand side of the figure: we observe that, even locally, there is a tight linear fit on the $500$ examples with highest privacy score. Additionally, $4,974$ of the $5,000$ examples that are easiest-to-attack are identical when computed on each of the two datasets.

## 4.2 The Onion Effect Is Not Explained by the Modified Dataset Being Smaller

While the CIFAR-10 dataset contains $50,000$ examples, our second dataset with the easiest-to-attack examples removed is just $45,000$ examples. In principle it is possible that the dataset being $10\%$ smaller puts each training sample at a much higher risk of privacy leakage.

In order to refute this hypothesis, we perform an experiment where we remove the same *number* of examples, but this time remove them from a different part of the data distribution. In particular, instead of removing the $5,000$ least private examples that are easiest to attack, we either remove the $5,000$ examples that are *most private* (i.e., hardest to attack) or we remove $5,000$ *random examples*.

Figure 5 plots the ROC curve for this analysis, and refutes the hypothesis that the Onion Effect is due to a reduction in dataset size. In fact, both our alternative configurations (removing the hardest examples, or random examples) result in nearly identical ROC curves as the baseline attack on the full dataset. The idealized ROC curve when the attack is prevented from attacking the hardest-to-attack samples is completely unchanged. This makes sense because the hardest-to-attack samples do not

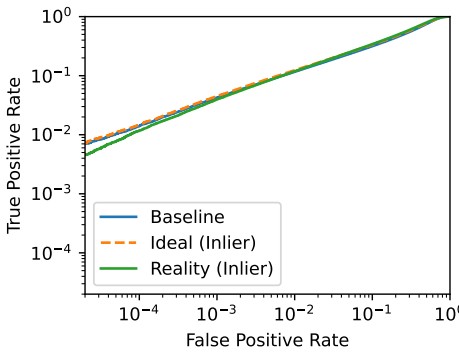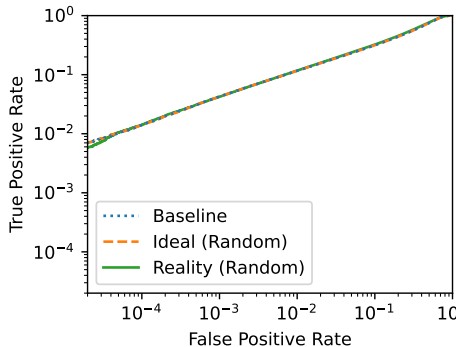

Figure 5: The Onion Effect can not be replicated when removing either the hardest-to-attack samples (left) or random samples (right).

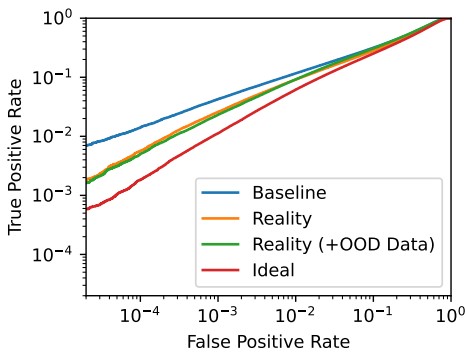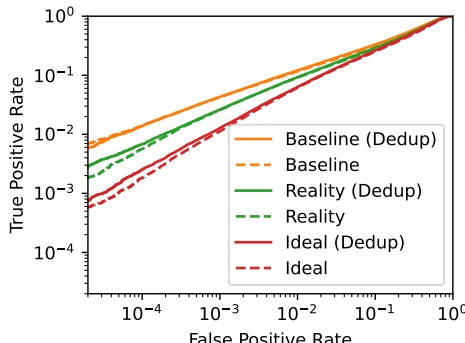

Figure 6: The Onion Effect remains even for datasets with artificially injected out-of-distribution outliers. We remove the 5,000 most vulnerable outliers from CIFAR-10 and insert 5,000 new randomly labeled outliers from CIFAR-100.

Figure 7: The Onion Effect still holds after deduplicating CIFAR-10 and removing 5,275 duplicated images from the entire dataset.

significantly contribute to the success rate at a low false-positive rate [6]. When we remove these examples and retrain the model, the experimentally observed results match: the attack remains exactly as effective as expected in the idealized model. Similarly, when we randomly remove 5,000 examples the attack success rate remains nearly identical. Taken together, these two experiments suggest that the Onion Effect is not be explained by a change in the size of the dataset.

## 4.3 The Onion Effect Is Not Explained by Limited Model Capacity

One potential explanation for the observed Onion Effect is that the model has limited capacity to memorize examples, and removing memorized examples frees this capacity up for new points to be memorized. Here, we refute this explanation, by demonstrating that arbitrary outliers do not produce the Onion Effect. To do so, we take the CIFAR-10 dataset, remove the 5,000 most outlier samples as we have done before, but then add back in 5,000 new outliers from CIFAR-100 with randomly given labels. Despite these examples being extreme outliers (because the inputs are out-of-distribution and the labels random), we find in Figure 6 that training on this augmented dataset does not alter the ROC curve away from the original curve in a statistically significant manner. That is, the Reality (+ OOD Data) curve in Figure 6 and the Idealized ROC curves both measure attack success rate on the same 45,000 examples, when 5,000 outliers are also present in the training set. However, the outliers from CIFAR-10 have a high impact on the attack success rate for the remaining 45,000 examples, while the CIFAR-100 outliers do not. This demonstrates that the specific outliers *do* matter, and that this effect cannot be explained by the existence of arbitrary outliers consuming model capacity.

### 4.4 The Onion Effect Is Not Explained by Duplicates in the Training Dataset

The CIFAR-10 dataset has many duplicates [26]. If an example is repeated many times, it is difficult to perform membership inference on any single one of these duplicates. Deduplicating the dataset will therefore increase many examples' membership inference accuracy, and might alter the results. Here we validate that this does not explain the Onion Effect. To do this we construct a deduplicated version of the CIFAR-10 dataset by using the open source image deduplication library `imagededup` [17] to remove 5,275 duplicated examples from the CIFAR-10 training set (see Appendix C for details). We then repeat our basic experiment on this deduplicated CIFAR-10 dataset and present the results in Figure 7. Interestingly, deduplication results in some examples having significantly higher attack success rate, as their duplicates in the training set "mask" their contribution.

## 5 Understanding the Onion Effect

We have suggested that the Onion Effect is a result of inliers becoming outliers when more extreme outliers are removed. In this section we provide experimental evidence for this claim, and then discuss the consequences for machine unlearning.

### 5.1 Transforming Inliers to Outliers

In order to demonstrate that the Onion Effect is in part explained by the previous inliers becoming outliers, we perform an experiment where we take previous samples where LiRA fails and remove just a few other (nearby) training examples to make the attack succeed. In particular, to target a example $x'$, we compute the influence of removing each training example $x$ on the privacy of $x'$ as:

$$\text{PRIVINF}(\text{REMOVE}(x) \Rightarrow x') \coloneqq \mathbb{E}_{f_s \leftarrow \mathcal{T}(X_s), X_s \leftarrow \mathbb{D}_X} \left[ \mathcal{A}(x', f_s) = \mathbb{1}[x' \in X_s] \mid x \notin X_s \right],$$

That is, we measure the membership inference accuracy on $x'$, averaged over all models trained without $x$. This is a natural extension of the definition of counterfactual influence proposed by [36] and similarly considered in [15].[1] We can compute PRIVINF for all training example simultaneously with the same random partitioning of the dataset used for LiRA. After computing these scores, we remove the $k$ training samples with the highest influence on our desired target. We focus our evaluation of this algorithm on four representative sets of examples, because we find the influence effect is not uniform across all examples.

**Duplicates.** Recall from Section 4.4 that deduplication increases many points' attack success rate, as duplicate points "mask" each other according to membership inference attacks. We consider the 5 points whose attack success rate increases the most after deduplication. Due to the masking effect, we expect the points with highest PRIVINF scores to be duplicates of these points. Indeed, the duplicates are the first points removed, and the removed points explain the entire attack improvement from deduplication. That is, for these points, the PRIVINF score identifies 1 or 2 duplicates which, when removed, increase membership inference advantage on the targeted points by an average of 42 percentage points.

**Second onion layer.** To validate that PRIVINF scores are not just a good deduplication tool, we now consider the 10 points with the highest increase in attack advantage after removing the 5,000 most vulnerable points in training (the initial onion experiment)—but excluding those with duplicates in the training set. If we remove all 5,000 most vulnerable points (the onion's first layer), the attack advantage on our 10 target points increases by 0.38 on average (from 0.42 to 0.8). For each of these 10 points, we find that instead removing *only the 25 points* with highest PRIVINF scores similarly increases membership inference advantage by 0.22 (i.e., 56% of the increase seen after removing all 5,000 points). Thus, **we find that the Onion Effect is a local effect rather than a global effect**: individual "second-layer" outliers are masked by only a small number of first-layer outliers. We show one example of a target image along with the first-layer outliers that mask it in Figure 8. This actually helps us better interpret our results from Section 4.3 where we injected out-of-distribution CIFAR-100 images to the CIFAR-10 raining set: CIFAR-100 examples are not "close enough" to the CIFAR-10 points to mask them, leading to very little impact on their vulnerability.

---

[1]Counterfactual memorization also includes a term for subsets with $x \in S$, but we find including this term makes our influence estimate slightly worse.

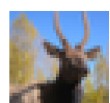 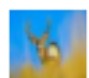 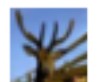 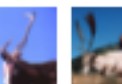 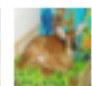 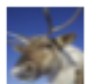 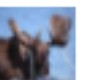 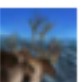 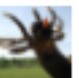 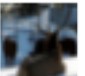

(a) Target                                      (b) Removed Samples

Figure 8: Removing 10 carefully chosen examples (drawn on the right) from the model's training set increases the membership inference accuracy of the target (shown on the left) from 66% to 81%.

**Random points.** Having validated that the removal of a few outliers could transform *some* points into new outliers, we now show that this effect is indeed limited to a "second layer" of outliers, and does not hold uniformly across the training set. Specifically, we find that if we target 10 *randomly chosen points*, then removing the 25 points with highest PRIVINF scores (for each target) only increases membership inference advantage by 0.08 of average (from 0.34 to 0.42).

**Safe points.** Finally, we study whether we can target "safe" points, which we define as points with (initially) lower than 2% membership inference advantage. Prior work has shown that it is possible to add poisoning data to a model to turn such points into outliers [33]. Here, we investigate whether *removing* data points can also turn the initially safest points into vulnerable outliers. For 10 targeted safe points, we find that removing the 25 points with highest PRIVINF scores (for each target) increases membership inference advantage by 0.12 on average (from 0.02 to 0.14). Yet, for some initially safe points, the attack advantage grows as high as 0.22 (an increase of 20 percentage points), nearly as high as for the (non-duplicate) points that are most impacted by the Onion Effect.

## 5.2   Consequences for Machine Unlearning

Privacy attacks like membership inference are commonly employed as primitives to measure progress in machine unlearning [3, 13], where a model needs to forget all it has learned from a training point, e.g., due to legislation promoting the "right-to-be-forgotten". The results we presented above have several implications for machine unlearning.

First, our results corroborate prior findings [31] which show that metrics based on membership inference should not be relied upon to measure, i.e., audit, how well a model has unlearned a point. If we did so, we might (erroneously) conclude that some points do not require to be actively unlearned, as the membership inference attack has negligible advantage on them. This could influence an individual user not to request their data to be unlearned; alternatively, the model owner might also decide that certain unlearning requests do not need to be actively acted upon. However, given the Onion Effect, a data point that is currently safe from membership inference could later become vulnerable if the underlying dataset changes. This could be the case, for instance, if other users make unlearning requests. This leads to a contradiction: the point initially believed to be unlearned (because membership inference attacks fail on it) would later be deemed to not be properly unlearned.

The experiment above with PRIVINF scores also suggests a form of *adversarial unlearning*: given a target individual's training point, an attacker could adaptively select other training points to be unlearned, to maximize the success of membership inference on the targeted individual.

## 6   Conclusion

Our experiments have several consequences for applied machine learning privacy.

**Membership inference attacks only audit specific (dataset, model) tuples.** In order to answer the question "is this model private?", researchers have suggested running empirical auditing analyses based on membership inference [23]. Our work suggests this approach may be flawed unless performed carefully. The results of any individual privacy audit should be restricted to the *exact* specific dataset that the model was trained on, and should be seen as a stable audit under any changes to this dataset. Thus, unless we expect the training set to never change, such audit results could mislead practitioners into believing that a model is private when, in fact, it may not be in the future on a different dataset. Interestingly, removing examples does not need to cause the Onion Effect, increasing attack success rate on remaining points. Removing examples can cause the reverse effect,

decreasing attack success rates, if the removed examples are poisoning examples as proposed by [33], or no effect at all, as in the arbitrary outlier experiments from Section 4.3. It is an interesting question for future work to investigate which effect dominates when removing examples in other settings.

**Instance-specific privacy-enhancing strategies.** At present, the dominant strategy to prevent memorization of training data is to strictly modify the *training algorithm* (e.g., to guarantee differential privacy). Our work can be seen as studying a potential alternative: modifying the training dataset.

While the Onion Effect suggests that defenses relying on *removing* examples from a training dataset will be ineffective, it stands to reason that *inserting* more extreme outliers might be able to make the original outliers more private. Unfortunately, the negative result from Section 4.3 suggests that adding arbitrary out of distribution data will not be effective. Alternatively, the results from Section 4.4 might seem to suggest that *duplicating* data points could help improve these points' privacy, but this reasoning is unfortunately also incorrect. Because a duplicate would be included in the training set if and only if the original example was in the training set, duplication actually makes membership inference *easier*: the adversary now needs to distinguish between models with 0 and 2 copies of a given training example, rather than between 0 and 1. Nevertheless we believe it is an interesting open question whether it would be possible to modify a training dataset to improve privacy.

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

## A  One-Shot vs Iterative Removal

If the definition of outlier depends on the other examples that are present, it is possible that removing 5,000 outliers in one pass might somehow impact the results—because we might miss some new set of examples that become outliers only after we have trained the model for a subset of epochs.

Here, we experiment with an iterative removal approach. In particular, starting with the 50,000 example dataset, we first run LiRA to find the easiest to attack samples then remove just the top 100 (instead of 5,000) examples. We then repeat our experiment on the remaining 49,900 example dataset; this gives us a *new* set of 100 examples to be removed from this dataset, which we do, giving us a dataset of 49,800 examples. We repeat this procedure 50 times until we are left with a dataset of 45,000 examples. We then plot in Figure 9 the ROC curve when attacking this dataset, and find that the results are almost completely identical to our initial experiments where we directly removed 5,000 outliers in one shot. Upon investigation, we find that the reason this occurs is because over 80% of the examples removed by the 1-shot approach are also removed by the 50 shot layer-by-layer approach.

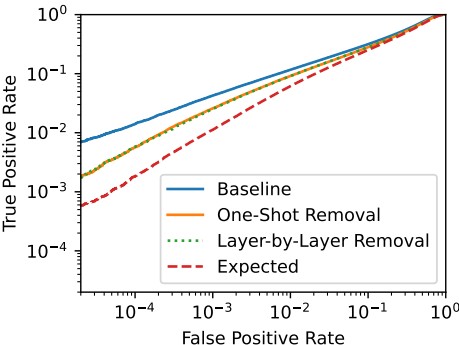

Figure 9: The Onion Effect is not a result of the outliers changing after each removal step. We repeatedly iterate 50 steps of identifying outliers and removing the top-100 outliers. This ends up also removing 5,000 outliers, and performs identically to the baseline configuration where we remove all 5,000 outliers in one shot.

## B    Visualization of Easy-to-Attack and Hard-to-Attack Examples

Figure 10 visualizes the easy to attack examples from CIFAR-10 training set, according to their privacy scores. The examples that are vulnerable to membership influence attack are generally outliers memorized by the models. The results are consistent with previous work that identify outliers and memorized examples in neural network learning [e.g. 12, 18].

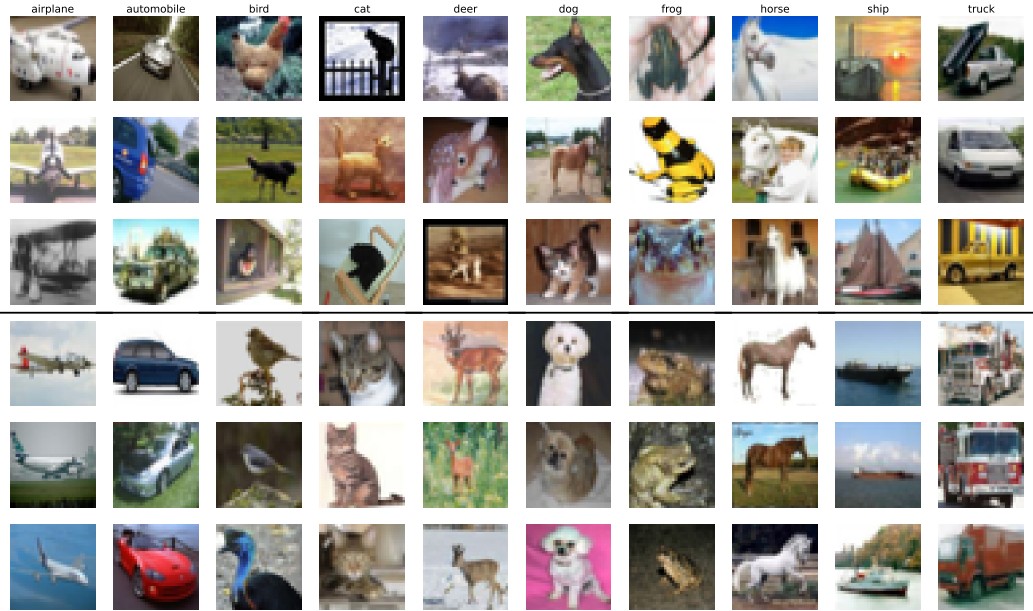

Figure 10: The easy to attack (top 3 rows) and difficult to attack (bottom 3 rows) CIFAR-10 examples from each of the 10 classes. Each column shows images from one class, with the top 3 rows randomly sampled from the top 100 most vulnerable examples, and the bottom 3 rows randomly sampled from the 100 least vulnerable examples.

# C    Details of CIFAR-10 Deduplication

We use the open source image deduplication library `imagededup` [17], available at `https://github.com/idealo/imagededup`, to deduplicate the CIFAR-10 training set. Specifically, we use the Convolutional Neural Network based detection algorithm with a threshold of 0.85, which ended up removing 5,275 duplicated images from the training set. We have manually inspected a random set of duplicated clusters identified by the software to verify the correctness. Figure 11 visualizes one of the largest duplicated clusters identified.

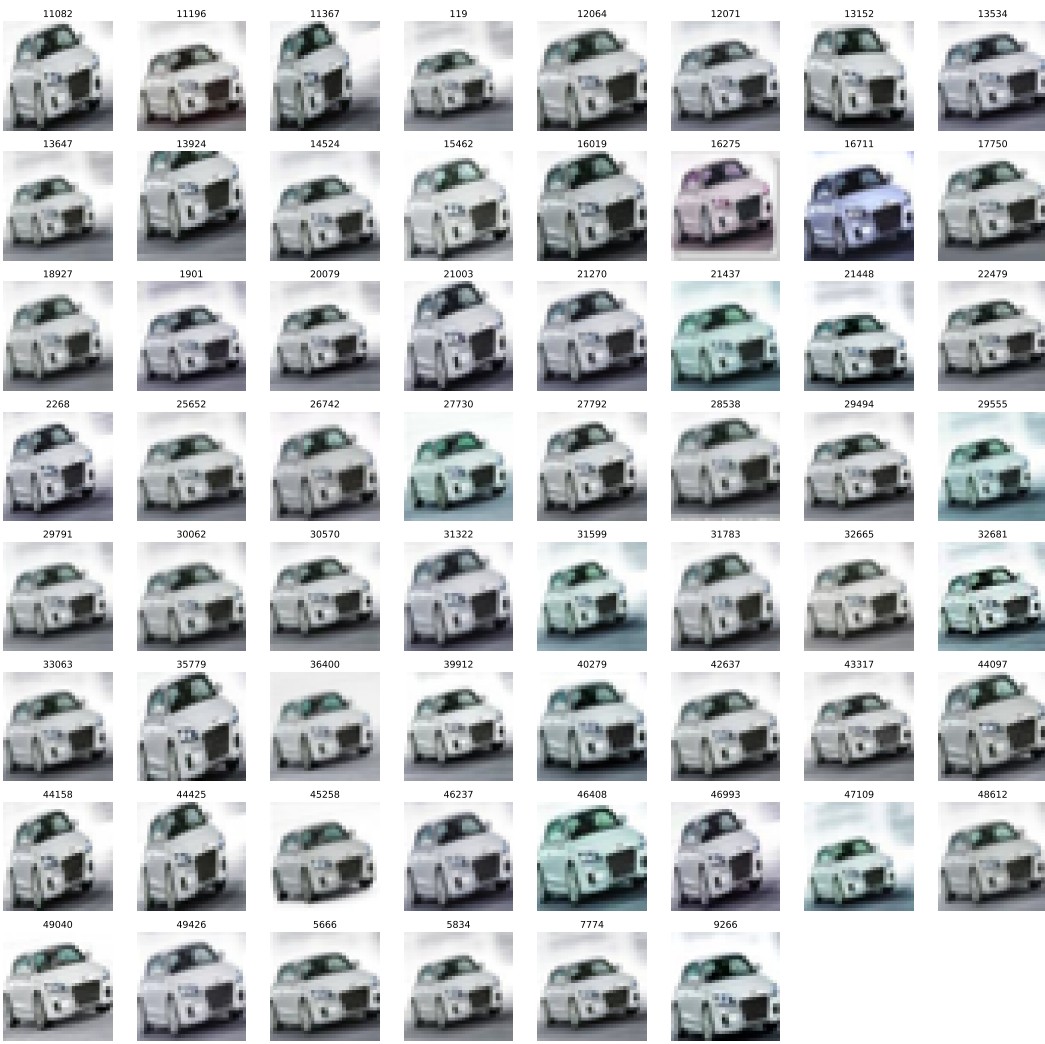

Figure 11: Duplicated images from the CIFAR-10 training set identified by the open source image deduplication library `imagededup` [17]. The number on top of each image is its index in the original CIFAR10 training set ordering.

