# OpenReview forum: "The Privacy Onion Effect: Memorization is Relative"
_NeurIPS.cc/2022/Conference — NeurIPS 2022 Accept_

### Official Review · Reviewer_EK8F · 2022-07-07

**Rating:** 7
**Confidence:** 4
**Soundness:** 3 good
**Presentation:** 4 excellent
**Contribution:** 3 good

**Summary:**

This paper shows removing easiest-to-attack examples makes some other examples easier-to-attack.

The authors define a privacy score to measure the difficulty of one example being successfully attacked. The privacy score is the average success rate of the Likelihood Ratio Attack (LiRA) on one example, which is computed over many models trained on subsets that contain the target example.

To demonstrate the Privacy Onion effect, the authors first remove some of the training data from CIFAR-10 and CIFAR-100 based on the privacy score. Then they re-compute privacy scores on the remaining data and find the LiRA membership inference attack achieves higher success rate on the remaining data.

The authors further discuss several intuitive hypotheses and how they fail to explain the Onion Effect. Finally, the authors show the Onion Effect is a local effect, i.e., removing a small number of carefully chosen examples can increase the attack success rate on a specific target example.


**Questions:**

1.Does the Privacy Onion Effect have infinite layers? Despite some of the examples experiencing higher privacy risks, the whole dataset becomes safer, i.e., the Reality line in Figure 1 is underneath the Ideal line. Now that removing 5000 examples improves the privacy by a factor of 2x, what if you remove 25000 examples? A plot showing the privacy improvement versus the number of removed datapoints would be nice.

2.Removing outliers may not necessarily transform inliers into outliers. In Section 5 you show the privacy risk of one example will increase if its ‘neighbor’ examples are removed (Figure 6). So, for the Privacy Onion Effect to occur, there is an implicit assumption that ‘outliers’ in the dataset also have neighbors. What if an outlier does not have any neighbors? CIFAR-10 has a relatively high data quality so the number of ‘pure outliers’ may be small.  If you remove a small number of datapoints from an extremely noisy dataset, e.g., large training corpus for big language models, will the Privacy Onion Effect still occur?

3.(Minor) You train 200000 models to ensure statistical validity. Is such a large number of models necessary? What will the results be if you only use 1000/5000/10000 models? Provide some discussion on this will ease the workload of reproducing the results and save unnecessary computations.

**Limitations:**

Please see the questions.

**Strengths And Weaknesses:**

**Strength:**

1.This paper is very well written. I enjoy reading it.

2.The findings in this paper are new and important to the community. The Onion Effect shows it is necessary to re-evaluate the privacy risk when training data changes, which guides future works on data removal for privacy protection. It also poses new challenges for machine unlearning.

3.The authors explore several possible explanations for the Onion Effect. The analysis further demonstrates the Onion Effect is a non-trivial finding and will help the community to better understand this problem.

4.Findings in this paper are supported by solid experiments, which are presented in an easy-to-understand fashion.

**Weaknesses:**

Please see the questions below.

---

> ### Author Response · Authors · 2022-08-02
> **Author Response**
>
> > Infinite layers?
>
> Removal at 25000 does appear to be somewhat more private than the original model/smaller removal counts, but is still very nonprivate (e.g. ~95% precision @fpr=10^-3). In an updated version of the paper we will include a figure showing this.
>
> > Pure outliers?
>
> This is an interesting question. For some pure outliers, it is likely that membership inference would already perform well, and so the onion effect may not be too extreme because there is not much room for attack improvement. Extending our findings to the large language model setting is an interesting direction for future work.
>
> >200000 models
>
> It is not necessary to train this many models. This effect is observable after training 256 models, which would only need ~1 GPU hour of compute. However, a criticism we received earlier was that this effect could just be due to random chance alone, and so we trained many (many!) more models to make sure this was not the case. And here we can see it is not the case.

---

> > ### Comment · Reviewer_EK8F · 2022-08-05
> > **Response**
> >
> > Thank you for your response. I'm a bit confused by your statement ''so the onion effect may not be too extreme because there is not much room for attack improvement''. My point is that removing some extreme outliers (say pure Gaussian noises) may not transfer any other datapoints into outliers, and hence the onion effect will not appear. I do not understand how this connects to the high attack success rate before data removal.
> >
> > In my humble opinion, the authors may need to mention that whether the onion effect will occur when data removal produces no new outliers is an open problem, i.e., the privacy onion effect may not always happen. Here is one evidence. In [1], the authors show that one can add some poisoned datapoints to turn some other datapoints into outliers. In that case, data removal may even transform outliners into inliers.
> >
> >
> > [1]: Truth Serum: Poisoning Machine Learning Models to Reveal Their Secrets, https://arxiv.org/pdf/2204.00032.pdf.

---

> > > ### Author Response · Authors · 2022-08-05
> > > **Thank you for clarifying!**
> > >
> > > Thank you for the comment. It seems we misinterpreted your original question.
> > >
> > > Your intuition is correct that there exist points whose removal does not change significantly other points' membership inference risk. In fact, our experiment in Section 4.3 can be seen as an experiment showing this - removing the very out-of-distribution CIFAR-100 examples (the extreme outliers) does not change membership risk for the other CIFAR-10 examples.
> > >
> > > We do mention Truth Serum in our paper, but the reviewer is correct that our submission lacks a single place directly comparing the effects of data removal observed in Section 5 (MI risk increases), Section 4.3 (MI risk doesn't change), and Truth Serum (MI risk decreases). We will add such a discussion, and pose as an open question which effect is most prominent in other settings mentioned by reviewers, such as text data and fine tuning.

---

### Official Review · Reviewer_8q8r · 2022-07-10

**Rating:** 7
**Confidence:** 4
**Soundness:** 4 excellent
**Presentation:** 4 excellent
**Contribution:** 2 fair

**Summary:**

This paper empirically demonstrates an effect they name 'onion effect' on classification for CIFAR-10 and CIFAR-100 datasets, where removal of the 'outliers' (top most *exposed* training data samples) results in increased memorization of other next layer exposed samples which is unlike what would be expected, a higher overall decrease in memorization. In this paper exposure/memorization is measured through a likelihood ratio based membership inference attack. The authors do extensive analysis to pinpoint/cross out possible reasons behind this phenomenon, and demonstrate that one possible explanation could be the fact that removing most exposed samples changes the 'support vectors' from those samples to some other closer ones, thereby making them more exposed. They also demonstrate that noise, model capacity, dataset size and duplicates aren't the reason behind this phenomenon.

**Questions:**


1. The experiments seem very compute intensive, specially given the number of models that need to be trained (1000 GPU hours), are there any easier/more efficient approximates of the likelihood ratio attack that can be run which are more efficient? The main reason behind this question is that this paper covers the onion effect for CIFAR classification task, and verifying this for larger transformer language models would need much more compute.


2. I wonder if this effect still exists on tasks other than classification (or classification with more labels), like language modeling or other generative models, as it seems like one important reason behind it is the discriminative nature and how the decision boundary (in the SVM example) falls on certain points and then moves with samples being removed. I wonder if this effect would still be so strong in different tasks, as there are more labels (like in language modeling).

3. As mentioned above, I am curious if the authors tested their hypothesis on pre-trained models too to see if the same thing happens there.

**Limitations:**

The only 'limitation' I see is the compute intensiveness of the attack, which is mentioned in the paper too. The authors do adequately discuss the consequences of the effect they have discovered, and possible ways to address it.

**Strengths And Weaknesses:**

Strengths:
1. The phenomenon that the authors have observed is new (to the best of of my knowledge) and  relevant,  as sample deletion and model unlearning are becoming concerns with new regulations.

2. The experimentation is thorough for the given task, as in they have covered all the reasons that would come to mind for why this happens and experimented with them. The influence metric is well designed to test out their hypothesis. The story that the paper tells is very coherent and the narrative is very easy to follow.


Weaknesses:

1.  The paper only looks at a simple classification task. Although this is a good proof of concept, it leaves me wondering if the phenomenon happens in other scenarios.

2. I think the effect of 'pre-training' is overlooked here. I understand that it might be outside the scope of the paper, but as the pre-train fine-tune paradigm is being applied more, I think seeing if this effect happens there as well is important. My guess is the pre-training could add many more samples to the 'SVM' (in the example), and then removing samples might not really move the boundary much.

---

> ### Author Response · Authors · 2022-08-02
> **Author Reply**
>
> > Fine tuning?
>
> This is an interesting question that is out of scope for this work. This is because there is little understanding of privacy leakage from finetuned models. A prerequisite to studying the onion effect on finetuning would be to adapt membership inference to finetuning.
>
> > Compute intensive?
>
> It is not necessary to train this many models. This effect is observable after training 256 models, which would only need ~1 GPU hour of compute. However, a criticism we received earlier was that this effect could just be due to random chance alone, and so we trained many (many!) more models to make sure this was not the case. And here we can see it is not the case.
>
> > Other tasks?
>
> This is another interesting question for future work. The point of our paper is to show that this effect exists in the first place, and we hope future work will replicate our results in other domains as well.

---

### Official Review · Reviewer_Mmkb · 2022-07-11

**Rating:** 6
**Confidence:** 3
**Soundness:** 3 good
**Presentation:** 4 excellent
**Contribution:** 3 good

**Summary:**

The paper addresses the role of relative memorizations on 'outlier data' in adversarial learning (membership inference attack).
Specifically, the authors first empirical demonstrate that removing the outlier points that are most vulnerable to a membership inference attack does not bring adequate improvement on the performance. Next, the authors investigate this 'Onion Effect' and provide some heuristics explanation. Finally, they provide several suggestions for current learning frameworks based on the 'Onion Effect'.

**Questions:**

Please see the comments in the Weakness/potential improvements  section above.

**Limitations:**

Please see the comments in the Weakness/potential improvements  section above.

**Strengths And Weaknesses:**

Merits:

- This paper is well written and organized.
- The concept of 'Onion Effect' is interesting.
- The implications towards current privacy mechanism are interesting and useful.




Weakness/potential improvements

- Where are the precise characterizations/derivations of those important factors mentioned in Lines  38 - 41?
All the experimental results used for comparison in the later sections are based on those factors. Please explicitly specify them.

- What is the intuition of the score function used in the paper？ Will any other score function lead to similarly results as observed in the paper?

- Further evaluations on other types of data, e.g., text

---

> ### Author Response · Authors · 2022-08-02
> **Author Reply**
>
> > Important factors?
>
> Here, what we mean is that, if points’ membership inference success rates were independent of each other, we would expect privacy to improve significantly. The privacy onion effect shows that their success rates are not independent of each other. We can change the text to make this more clear.
>
> > Score function?
>
> We use the LiRA score function, as it is the current state of the art membership inference attack. See our response to reviewer t45T for more justification.
>
> > Other data types?
>
> In order to train many models, we are limited to datasets where training is fast. So we run our experiments mostly on CIFAR-10 and CIFAR-100 where we can train models in ~dozens of seconds.

---

> > ### Comment · Reviewer_Mmkb · 2022-08-09
> > **Thanks for the clarifications**
> >
> > Thank you for your responses. I will keep my positive score.

---

### Official Review · Reviewer_t45T · 2022-07-11

**Rating:** 6
**Confidence:** 4
**Soundness:** 3 good
**Presentation:** 4 excellent
**Contribution:** 3 good

**Summary:**

This paper examines the impact of removing records that are most vulnerable to a membership inference attack on the privacy of the remaining records. Given a machine learning model trained on a private dataset, one way to make it more “privacy-preserving” without randomizing the training algorithm could be to identify records at risk, remove them from the dataset, then re-train a model on the remaining records. The paper shows that some of the remaining records become more at risk and investigates potential causes for this phenomenon.

**Questions:**

I believe further analysis is needed, for instance reporting in the paper/appendix the following:
- Summary statistics (e.g. histogram) of the Attack Success Rates for the 50,000 records.
- Error bars for the ASR estimates (if applicable).
- Overall attack accuracy/AUC. Can the Onion effect also be noticed using coarser metrics (e.g. accuracy) rather than TPR for low FPR?
- An explanation for the necessity of training 200,000 models - this is a great computational cost, orders of magnitude above what is used in previous work [6] (256 models). I understand this to be the main reason for not extending the experiments to other settings; why is it needed?

The paper could define more precisely inliers/outliers and how they relate to “easy-to-attack” and “hard-to-attack”. The current usage of the term is quite confusing, e.g., “inliers becoming outliers when more extreme outliers are removed” (lines 259-260).

Minor points:
- Why is the Duplicate analysis (line 274) carried out on 5 points while the others on 10 points? Can the analyses be extended to more points?
- Fig. 4 (left) and the corresponding paragraph (241-247) are hard to understand. Why are only partial results shown, i.e. why are the Baseline (45,000 + OOD) and Ideal (45,000 + OOD) results omitted? It seems to me (but I could be wrong) that the text explaining the figure contradicts the results.
- On Fig. 3 (left), why is Reality (inlier) lower than the others? Is the gap statistically significant? The text does not seem to acknowledge the gap.
- Using the same scale on Fig. 1 for CIFAR10 and CIFAR100 would make it easier to visually compare the gaps.
- What criterion guides the choice of 5000 (and not, say 1000 or 500) vulnerable records to be removed? Is there some statistical reason for this?
- In the legend of Figure 7, does “Expected” correspond to the main paper’s “Ideal”? If so, it would help to make the naming consistent.
- The use of consistent colors across the different figures, e.g., blue, orange, and green to denote the same setup, would make it easier to follow the paper and to compare the results of different figures.
- Will the authors release the 200,000 shadow models? This could be a useful resource for the community and would allow to replicate the results.

**Limitations:**

The authors make relevant recommendations to practitioners.

**Strengths And Weaknesses:**

Strengths:
- The paper takes the study of the impact of outliers on MIA a step further by identifying, measuring, and studying potential causes of the “privacy onion” effect.
- The study of potential causes is principled and well-explained.
- Relevant recommendations are made to practitioners.

Weaknesses:
- The causal analysis is limited to one dataset (CIFAR10), and all the analyses are limited to one attack and one target model. To support the paper’s broad, general claims, I believe further evidence is needed that this is indeed a general phenomenon. The same applies to the causes identified on CIFAR10, LiRA [6], and one specific target model: are these insights applicable to other settings? In particular, the outlier analysis (Sec. 4.3) is quite surprising and it would be useful to replicate the finding using different outliers or a different dataset.
- Lack of reproducibility: important details are missing from the paper, such as the target model architecture and (train/test) accuracy, the number of training samples for fitting the Gaussians vs number of test samples for plotting the ROC curves, etc.

---

> ### Author Response · Authors · 2022-08-02
> **Author Reply**
>
> We thank the reviewer for their detailed questions.
>
> > “Limited to one dataset” / “Limited to one target model”
>
> In Figure 1 (right) we demonstrate this effect holds on CIFAR-100 as well. In preliminary analysis we also confirmed this effect held true on MNIST with a different CNN architecture.
>
> > “limited to one attack”
>
> All membership inference attacks are essentially the same (Sablayrolles et al. 2019); we just happen to use the current state-of-the-art technique. Also note that membership inference attacks form the basis of all other privacy attacks (e.g., training data extraction).
>
> > “Lack of reproducibility”
>
> We will add additional details as requested by the reviewer, but please note that we run an open-source training algorithm out-of-the-box without modification, and so anyone should be able to reproduce our results.
>
> > “Error bars for ASR estimates”
>
> We trained 200k models, and so our error bars are <0.01% and would be essentially indistinguishable from the mean.
>
> > “Overall attack accuracy/AUC. Can the Onion effect also be noticed using coarser metrics (e.g. accuracy) rather than TPR for low FPR?”
>
> This is an interesting question; we will investigate this for an updated version of the paper. For now, we believe that (as is argued in prior work) AUC is not a good metric for privacy and so even if AUC was impacted this would not matter much.
>
> >  “Necessity of training 200k models”
>
> It is not necessary at all. This effect is observable after training 256 models, which would only need ~1 GPU hour of compute. However, a criticism we received earlier was that this effect could just be due to random chance alone, and so we trained many (many!) more models to make sure this was not the case. And here we can see it is not the case.
>
> > “Is duplicate analysis different on top 10?”
> We reran this analysis on the top 10 duplicates and find the results are consistent. Advantage increases by 38 percentage points on top 10, vs 42 on top 5.

---

> > ### Comment · Reviewer_t45T · 2022-08-09
> > **Response**
> >
> > Thank you for the reply.
> >
> > -    “Summary statistics (e.g. histogram) of the Attack Success Rates for the 50,000 records.”
> >
> > I am coming back to this point because I am not sure from reading the response if the authors will address it. I still think it would be useful to include in the paper a histogram or summary statistics of the Attack Success Rates. This would help assess, empirically, the criterion used to remove the records. I would like to see, for instance, if the ASRs of the top 10% records which are being removed are much larger, or quite similar, to the ASRs of the remaining records.

---

> > > ### Author Response · Authors · 2022-08-09
> > > **We can add a histogram**
> > >
> > > We can add a histogram like this to the appendix. In the meantime, here are some percentiles:
> > >
> > > min: 0.499
> > >
> > > 10%: 0.506
> > >
> > > 20%: 0.511
> > >
> > > 30%: 0.523
> > >
> > > 40%: 0.545
> > >
> > > median: 0.577
> > >
> > > 60%: 0.620
> > >
> > > 70%: 0.674
> > >
> > > 80%: 0.744
> > >
> > > 90%: 0.833
> > >
> > > max: 0.997
> > >
> > > There's a fairly smooth degradation in the ASR: the next 10% largest are in [74%, 83%], while the top 10% are in [83%, 100%]. Here are the percentiles after the removal (on unremoved points), for comparison:
> > >
> > > min: 0.486
> > >
> > > 10%: 0.506
> > >
> > > 20%: 0.511
> > >
> > > 30%: 0.517
> > >
> > > 40%: 0.530
> > >
> > > median: 0.555
> > >
> > > 60%: 0.593
> > >
> > > 70%: 0.644
> > >
> > > 80%: 0.712
> > >
> > > 90%: 0.797
> > >
> > > max: 0.994

---

### Meta-Review · Area_Chair_uSFv · 2022-08-23

**Recommendation:** Accept
**Confidence:** Certain

**Metareview:**

Reviewers found the paper to be thought provoking and relevant and would be of broad interest to the community.


**Award:**

No

---

### Decision · Program_Chairs · 2022-09-14

Accept